# A Hybrid Approach Combining Shape-Based and Docking Methods to Identify Novel Potential P2X7 Antagonists from Natural Product Databases

**DOI:** 10.3390/ph17050592

**Published:** 2024-05-07

**Authors:** Natiele Carla da Silva Ferreira, Lucas Gasparello Viviani, Lauro Miranda Lima, Antonia Tavares do Amaral, João Victor Paiva Romano, Anderson Lage Fortunato, Rafael Ferreira Soares, Anael Viana Pinto Alberto, Jose Aguiar Coelho Neto, Luiz Anastacio Alves

**Affiliations:** 1Laboratory of Cellular Communication, Oswaldo Cruz Foundation, Rio de Janeiro 21040-360, Brazil; natielec15@gmail.com (N.C.d.S.F.); lauro.fma@gmail.com (L.M.L.); jromano@micro.ufrj.br (J.V.P.R.); andersonlage2043@gmail.com (A.L.F.); aanael@gmail.com (A.V.P.A.); 2Institute of Chemistry, University of São Paulo, São Paulo 05508-000, Brazil; lucas.viviani@usp.br (L.G.V.); atdamara@iq.usp.br (A.T.d.A.); 3Laboratory of Immunobiotechnology, Federal University of Rio de Janeiro, Rio de Janeiro 21941-590, Brazil; 4Laboratory of Applied Genomics and Bioinnovations, Oswaldo Cruz Foundation, Rio de Janeiro 21040-360, Brazil; rafael.ferreira.fiocruz@gmail.com; 5National Institute of Industrial Property, Rio de Janeiro 20090-910, Brazil; jacnet62@gmail.com; 6Tijuca Campus, Veiga de Almeida University, Rio de Janeiro 20271-020, Brazil

**Keywords:** P2X7 receptor, antagonists, natural products, virtual screening, shape-based model

## Abstract

P2X7 is an ATP-activated purinergic receptor implicated in pro-inflammatory responses. It is associated with the development of several diseases, including inflammatory and neurodegenerative conditions. Although several P2X7 receptor antagonists have recently been reported in the literature, none of them is approved for clinical use. However, the structure of the known antagonists can serve as a scaffold for discovering effective compounds in clinical therapy. This study aimed to propose an improved virtual screening methodology for the identification of novel potential P2X7 receptor antagonists from natural products through the combination of shape-based and docking approaches. First, a shape-based screening was performed based on the structure of JNJ-47965567, a P2X7 antagonist, using two natural product compound databases, MEGx (~5.8 × 10^3^ compounds) and NATx (~32 × 10^3^ compounds). Then, the compounds selected by the proposed shape-based model, with Shape–Tanimoto score values ranging between 0.624 and 0.799, were filtered for drug-like properties. Finally, the compounds that met the drug-like filter criteria were docked into the P2X7 allosteric binding site, using the docking programs *GOLD* and *DockThor*. The docking poses with the best score values were submitted to careful visual inspection of the P2X7 allosteric binding site. Based on our established visual inspection criteria, four compounds from the MEGx database and four from the NATx database were finally selected as potential P2X7 receptor antagonists. The selected compounds are structurally different from known P2X7 antagonists, have drug-like properties, and are predicted to interact with key P2X7 allosteric binding pocket residues, including F88, F92, F95, F103, M105, F108, Y295, Y298, and I310. Therefore, the combination of shape-based screening and docking approaches proposed in our study has proven useful in selecting potential novel P2X7 antagonist candidates from natural-product-derived compounds databases. This approach could also be useful for selecting potential inhibitors/antagonists of other receptors and/or biological targets.

## 1. Introduction

P2X7 is a member of the P2X purinergic receptor family, which is physiologically activated by extracellular ATP [1]. P2X7 activation results in the opening of a cation-nonselective channel that allows for the flux of Ca^2+^, Na^+^, and K^+^ ions through the cell membrane, according to the electrical and concentration gradient. Prolonged exposure to high ATP concentrations (>100 μM) induces the formation of a reversible membrane pore, which allows for the passage of molecules of up to 900 Da [2]. P2X7 activation also promotes a series of pro-inflammatory responses, including caspase activation, cytokine release, and reactive oxygen species generation, in addition to cell death [3,4,5,6,7].

P2X7 is considered a promising therapeutic target for several diseases and pathophysiological events, such as cancer, pain, and neurodegenerative, inflammatory, and infectious diseases [8,9,10,11,12]. P2X7 antagonists have shown potent anti-inflammatory and antinociceptive effects in vivo, reinforcing their therapeutic importance in several pathophysiological contexts [5,13]. Currently, several compounds have been reported as presenting antagonistic activity against P2X7, and some of them were identified through high-throughput screening (HTS) campaigns carried out by pharmaceutical companies. Nevertheless, none of the known P2X7 antagonists have been approved for use in clinical therapy, due to pharmacokinetic limitations or the lack of clinical efficacy [5,14]. This motivates the search for novel P2X7 antagonists.

Natural products have emerged as a promising source for the discovery of novel drugs, due to a significant diversity of chemical structures presenting a potential bioactivity acquired over thousands of years of evolution [15]. Notably, almost 49% of all new chemical entities released between 1981 and 2019 are natural products, semi-synthetic derivatives, or synthetic drugs with a pharmacophore derived from natural products [16], highlighting their importance in drug discovery. Furthermore, over 10 compounds from plants, animals, and microorganisms have been identified with antagonistic or modulatory activity on P2X7 [17,18], including emodin [19,20,21], colchicine [22], and stylissadines [23].

Classical drug discovery strategies include in vitro or phenotypic assays in an HTS context [24] and/or in silico methods [25]. The HTS strategy is frequently used in the pharmaceutical industry as it allows for testing thousands of compounds through bioassays. This approach, however, is time-consuming, laborious, and costly [25]. In this regard, virtual screening (VS) has been well established as the main in silico approach to cheaply and quickly screen very large databases (10^6^–10^8^) to identify compounds with predicted biological activities [26,27,28]. Compounds selected by VS campaigns must, nevertheless, be submitted to confirmatory in vitro and/or in vivo assays for VS experimental validation [26,27,29,30].

Shape-based 3D similarity methods have been widely used as a VS approach [31,32,33], with several recently reported successful applications [30,34,35], including for the selection of inhibitors of the human ecto-5′-nucleotidase (h-ecto-5′-NT, CD73), an enzyme that plays a key role in purinergic signaling pathways [30]. Shape-based 3D similarity methods are based on the premise of a shape complementarity between a ligand and its corresponding binding site in the receptor [31,32,33]. Therefore, compounds with a similar shape would have a high probability of presenting the same biological activity(ies) against a certain biological target [31]. One of the main advantages of shape-based similarity methods is that they allow for the selection of structurally diverse compounds, despite presenting similar shapes [31,32].

Docking methods were originally proposed in the literature to predict the binding mode of a ligand to a target protein [36,37,38]. Docking is frequently used, alone or in combination with other in silico approaches, as part of VS protocols [36,37,38]. Docking consists of two major steps: (i) the prediction of ligand conformation and orientation inside the binding site of the target protein and (ii) the prediction of protein-ligand affinity, usually applying “scoring functions” [39]. Despite the recent advancements concerning the development of docking methods, the scoring functions currently available remain quite limited in their abilities to consider relevant protein–ligand interaction aspects, including solvation/desolvation, polarizability, and entropic effects [38,39,40]. Additionally, accurate predictions of binding modes and binding affinities are limited by (i) the inherent protein flexibility, (ii) the undersampling of ligand conformational states, (iii) induced fit effects or other conformational changes that may occur upon ligand binding, (iv) the resolution of crystallographic structures available for docking, and (v) the lack of a complete comprehension on the role of water molecules in protein–ligand interactions [38,39,40,41,42]. Due to these recognized limitations, a visual inspection of the predicted docking poses inside the protein binding site is crucial for docking-based VS approaches [30,36,40,43,44]. Additionally, experimental validation steps are extremely relevant to overcome inherent docking limitations [30].

Herein, we aimed to search for novel potential P2X7 antagonists using a VS approach consisting of (i) a shape-based filter proposed based on the structure of JNJ-47965567, a known P2X7 receptor antagonist that binds to an allosteric binding site in this receptor [45,46]; (ii) a drug-like filter; and (iii) a docking filter followed by a careful visual inspection of the docking poses into the P2X7 allosteric binding site. Each of these filters was applied to the following AnalytiCon natural product databases: MEGx (~5.8 × 10^3^ compounds) and NATx (~32 × 10^3^ compounds). Using this VS approach, four compounds from the MEGx database and four from NATx were selected as potential P2X7 antagonists, representing >99% reduction in the number of compounds from each database. The selected compounds represent structurally novel drug-like P2X7 antagonist candidates.

## 2. Results and Discussion

A schematic representation of the VS protocols applied herein to select potential natural-product-derived P2X7 antagonist candidates is shown in Figure 1. As mentioned above, the VS protocols were applied to the following AnalytiCon databases: MEGx (~5.8 × 10^3^ compounds), which encompasses isolated natural products from plants and microorganisms, and NATx (~32 × 10^3^ compounds), which comprises synthetic compounds derived from natural products.

### 2.1. Shape-Based Screening

In the first step of our VS protocol, a shape-based model was generated based on the structure of JNJ-47965567 (Figure 2). This compound has been described in the literature as a non-competitive and selective P2X7 antagonist, presenting inhibitory potency in the low nanomolar range (IC_50_ = 11.9 nM) [45,46]. The crystal structure of P2X7 from *Ailuropoda melanoleuca* (giant panda), referred to herein as *Am*P2X7, complexed with JNJ-47965567 (PDB ID: 5U1X) has been reported in the literature, revealing that JNJ-47965567 occupies an allosteric binding pocket located at the interface between two adjacent P2X7 subunits (Figure 3A,B) [46]. Notably, *Am*P2X7 has a high sequence identity (~85%) to the *Homo sapiens* P2X7 (*Hs*P2X7), and all the residues that form the allosteric binding site are conserved among *Am*P2X7 and *Hs*P2X7 (Appendix A). Therefore, the *Am*P2X7 crystal structure was chosen as the reference structure for our studies. *Am*P2X7 allosteric binding pocket has also been verified as accommodating other structurally diverse *Am*P2X7 antagonists, albeit with lower binding affinities, highlighting the crucial roles of the size and shape of this pocket in the molecular recognition of antagonists by *Am*P2X7 [46]. The relevance of shape complementarity for receptor-antagonist binding supports the choice of a shape-based model as the first filter in our VS protocol. Our shape-based model was built considering the same JNJ-47965567 conformation as observed in the crystal structure of the *Am*P2X7-JNJ-47965567 complex (Figure 3C) [46].

Subsequently, the generated shape-based model was applied to the MEGx and NATx databases. The top-ranked 202 and 1005 compounds from each database, respectively, were selected for the next step (Figure 1), representing ~3.5% and ~3.1% of the total number of compounds in these databases. The Shape-Tanimoto score values of the compounds from the MEGx and NATx databases ranged from 0.624 to 0.698 and from 0.679 to 0.799, respectively. To validate our shape-based approach, 79 compounds described in the literature as P2X7 allosteric antagonists were applied to the generated shape-based model. The Shape-Tanimoto score values of the known antagonists ranged from 0.467 to 0.694 (Appendix A). Therefore, the compounds selected from the MEGx and NATx databases as novel potential P2X7 antagonist candidates by our generated shape-based model have Shape-Tanimoto score values in the same range or even higher than the Shape-Tanimoto score values of P2X7 allosteric antagonists described in the literature and applied to the same model.

### 2.2. Drug-like Filter

Aiming to filter for potentially drug-like compounds, those selected by the shape-based screening in the first step of the VS protocol were subsequently submitted to a drug-like filter, using the “blockbuster” criteria implemented in the *FILTER* program v.4.1.2.0 [47,48]. As a result, 126 and 977 compounds from the MEGx and NATx databases (Figure 1), respectively, were retained for the next step, representing a ~37.6% and ~2.8% reduction in the number of compounds selected from both databases by the shape-based screening.

### 2.3. Docking and Visual Inspection

Compounds that met the applied drug-like filter criteria were finally docked into the *Am*P2X7 allosteric binding pocket (see Figure 3B), using the *GOLD* v.5.2 and *DockThor* v.2.0 programs. The docking procedures were validated by redocking JNJ-47965567 into the allosteric binding site. The redocking procedure was also performed for four other *Am*P2X7 antagonists that occupy the same allosteric binding site and whose *Am*P2X7R-complexed structures have been elucidated by X-ray crystallography and deposited into PDB (PDB IDs: 5U1U, 5U1V, 5U1W, and 5U1Y). Most of the protein–ligand interactions observed in the crystal structures were reproduced by the respective best ranked docking poses obtained using *GOLD* and *DockThor* software (Appendix A).

Next, 25 compounds from the MEGx database with the highest score values according to the *GOLD ChemPLP* scoring function (PLP score values ranging from 86.2 to 107.5) and 25 compounds from the MEGx database with the highest score values according to the *DockThor* scoring function (score values ranging from −12.2 to −10.6 kcal/mol) were selected for further analyses. Among the 50 compounds, 10 were simultaneously selected by both docking programs, which means that the total number of compounds selected from the MEGx database was 40 (~68.3% reduction in the number of docked compounds; Figure 1). Additionally, 100 compounds from the NATx database with the highest score values according to the *GOLD ChemPLP* scoring function (PLP score values ranging from 99.9 to 111.1) and 100 compounds from the NATx database with the highest score values according to the *DockThor* scoring function (score values ranging from −11.8 to −10.9 kcal/mol) were also selected for further analyses. As 18 compounds were simultaneously selected from both docking programs, the total number of compounds selected from the NATx database was 182 (~81.4% reduction in the number of docked compounds; Figure 1).

Aiming to select the compounds that greatest fit into the binding site, the best scoring pose of each compound was analyzed by visual inspection into the *Am*P2X7 allosteric binding site. To establish criteria that could aid in guiding our visual inspection, we carefully analyzed the protein–ligand interactions in the crystal structures of the five aforementioned *Am*P2X7-antagonist complexes, including *Am*P2X7-JNJ47965567 (see Appendix A). Our analysis revealed that at least three of the five analyzed antagonists interact with the following residues from the *Am*P2X7 allosteric binding site via hydrophobic interactions: F88, F95, M105, F108, Y295, and I310. Additionally, at least three establish a hydrogen bond and/or an ionic interaction with D92 and a hydrogen bond or a hydrophobic interaction with Y298 (from the adjacent monomer). The importance of the interactions with many of these residues for protein–ligand recognition has already been experimentally confirmed by site-directed mutagenesis studies, as reported in the literature [46]. Moreover, interaction with F103 is essential for the inhibitory activity of all antagonists [46].

Based on these *Am*P2X7-antagonist interaction analyses, we established that the compounds selected by our docking filter should fulfill the following criteria to be selected in the visual inspection analysis: (i) make at least six interactions with any allosteric binding site residues; (ii) make interactions with at least four “key-residues”, as follows: a hydrophobic interaction with F88, F95, F103, M105, F108, Y295, and/or I310; a hydrogen bond and/or an ionic interaction with D92 and/or a hydrogen bond or a hydrophobic interaction with Y298 (from the adjacent monomer); and (iii) make at least one hydrogen-bond interaction, ionic interaction, or cation–pi interaction with any other allosteric binding site residue. Additionally, the following criteria should also be fulfilled: (i) there should be a shape complementarity between the compound and the allosteric binding site; (ii) the docking pose with the best score should be reproducible; and (iii) the conformation of the docking pose with the best score should not be sterically hindered.

Compounds whose docking poses from both *GOLD* and *DockThor* met the visual inspection criteria were finally selected as potential *Am*P2X7 antagonists, encompassing four compounds from the MEGx database and four compounds from the NATx database (Figure 1). This represents 90.0% and ~97.9% reductions in the number of docked compounds from each database, respectively. The structures of selected compounds, some of their physicochemical properties, and their *GOLD ChemPLP* and *DockThor* score values are shown in Table 1. Notably, the *GOLD ChemPLP* score values for the compounds selected by our VS protocol (89.99 to 101.31) are overall higher than the score values of the five known aforementioned *Am*P2X7 antagonists (67.35 to 97.22), which were redocked into the *Am*P2X7 allosteric binding site (Appendix A). Additionally, the *DockThor* score values for the compounds selected by VS (−12.169 to −10.924) are lower than the score values of the five known *Am*P2X7 antagonists (−11.242 to −9.601; Appendix A). These results suggest that the compounds selected by our VS would have higher affinities to *Am*P2X7 in comparison to the known *Am*P2X7 antagonists. Nevertheless, as mentioned above, experimental validation is a unique way of confirming in silico predictions.

The main protein–ligand interactions between the best scoring docking poses of each selected compound from the MEGx and NATx databases and the residues that form the *Am*P2X7 allosteric binding site are shown in Table 2 and Table 3, respectively. Remarkably, all docking poses from the compounds from both databases make hydrophobic interactions with F95, which has been recognized as essential for the binding affinity of known P2X7 antagonists that bind the same allosteric pocket, as discussed above and reported in the literature [46]. Additionally, all compounds make a cation–pi or a hydrogen bond with K297 according to *GOLD* and/or *DockThor* predictions. Moreover, at least seven of the eight selected compounds make hydrophobic and/or π-stacking interactions with residues F88, Y295, and I310. All compounds from the NATx database make a hydrophobic interaction with F103, another residue that is recognized as crucial for the binding of known antagonists [46].

Interestingly, many of these residues are located near energetically favorable regions to position hydrophobic, hydrogen-bond donor, and hydrogen-bond acceptor groups, as revealed by the molecular interaction fields (MIFs) [49,50,51] calculated using the *GRID* program v.2021.3 [49,50,51] (Figure 4). An extensive hydrophobic region is observed near residues F95, F103, F293, and Y295. A hydrophobic contour is also observed near residues F95, F03, and Y295 from the adjacent subunit. Hydrogen-bond acceptor regions are detected close to K297.

Notably, a high overlap between *GOLD* and *DockThor* poses for compound NP-016468 from the MEGx database is observed, as displayed in Figure 5A. Hydrogen bonds with K110, A91, and Y298 (from the adjacent subunit) are observed for both these docking poses. Additionally, hydrophobic interactions with F88, F95, F293, and I310 are common to both docking poses (Table 2). Among the compounds selected from the NATx database, the best overlap between *GOLD* and *DockThor* docking poses was observed for NAT13-340161 (Figure 5B), displaying hydrogen bonds with K297 and with Y298 (from the adjacent subunit), and hydrophobic interactions with F88, F95, F103, and F293 (Table 3). 

### 2.4. Molecular Dynamics (MD) Simulations

Finally, we conducted MD simulations to evaluate the dynamic behavior of compounds with protein flexibility, since in docking, the protein is maintained rigidly. This test could allow more realistic protein–ligand interactions. As shown in Figure 6A, the compound NP-016468 is not sufficiently stabilized in the P2X7 binding site and an extended simulation time is necessary to observe this feature. Despite this, it is worth noting that within 50 ns, the compound did not dissociate from the binding site, suggesting that it is not likely to be ejected. The compound NP-025357 appears to stabilize its conformation in a short time, around ~40 ns, as observed in Figure 6B. Although an extended time for simulation is required for a more precise assessment, this behavior suggests that this compound might have a greater affinity for P2X7. On the other hand, with a longer period, it was possible to observe the stability of the compound NP-025047 in the P2X7 binding site. There are two main instances of ligand accommodation observed at 30 and 45 ns peaks (Figure 6C).

There is a predominance of hydrophobic and ring-stacking interactions such as π-stacking and π-π in the analyzed interactions, as previously observed in docking. As shown in Figure 7, compound NP-016468 mainly interacts with Phe, Tyr, and Trp residues. A similar pattern of interactions was observed with compound NP-025357 in Figure 8. Furthermore, it is possible to observe two hydrogen bonds occurring between the compound and the residues D92B and Y298C, which likely stabilized the ligand rapidly. It is important to note that an extended computational time would be necessary to reinforce these assertions. Finally, compound NP-025047 also presents a similar profile, involving P96 in addition to the previously described Phe, Tyr, and Trp residues (Figure 9). These multiple interactions between aromatic residues and the compound rings reinforced its stability over the MD simulation time. Nevertheless, more than 150 ns was necessary to notice stability, suggesting that the compound may not be ejected, due to the narrow exit of the binding site.

In summary, the combination of shape-based and docking approaches presented has proven useful for the selection of eight natural product-derived compounds as novel P2X7 antagonist candidates. The selected compounds are structurally different from known P2X7 antagonists, display drug-like properties, and are predicted to interact with key residues of the P2X7 allosteric binding pocket, including F88, F92, F95, F103, M105, F108, Y295, Y298, and I310. Finally, we point out that these compounds should be tested in in vitro assays for the experimental validation of the proposed VS protocol due to the inherent limitations of in silico methods.

## 3. Material and Methods

### 3.1. Compound Databases’ Preparation

The MEGx (~5.8 × 10^3^ compounds) and NATx (~32 × 10^3^ compounds) natural product databases from the AnalytiCon Discovery library (release 04/2022; available at https://ac-discovery.com/, (accessed on 2 January 2024)) were obtained in SDF format and used for the VS protocols in this study. The 3D coordinates of each compound from these databases were obtained using the *OpenBabel* v.3.1.1 software. The most abundant tautomeric form and the predominant protonation states of the ionizable groups for each compound at pH 7.4 were obtained using the *Tautomers* and *FixpKa* tools available in the *QUACPAC* v.2.1.3.0 program (OpenEye Scientific Software, Santa Fe, NM, USA). Subsequently, up to 50 conformations of each compound were generated using the *OMEGA* v.4.1.2.0 program (OpenEye Scientific Software, Santa Fe, NM, USA) [47,48], with default parameters.

### 3.2. Shape-Based Screening Procedures

The 3D structure of the JNJ-47965567 antagonist was obtained from the crystal structure of the P2X7R-JNJ-47965567 complex (PDB ID: 5U1X, resolution: 3.20 Å) and used for building a shape-based model (“query”) using the ROCS v.3.5.0.2 program (OpenEye Scientific Software, Santa Fe, NM, USA) [31] applying the default settings. The JNJ-47965567 conformation considered to build the model was the same as that observed in the crystal structure. The model was applied to the MEGx and NATx databases, which were prepared as described above. As reported in the literature, the *ROCS* software overlays candidate molecule conformation to the generated query based on their shape matches. The conformation that best matched the model for each compound in the MEGx and NATx databases was ranked according to the Shape-Tanimoto score value, which ranges from 0 to 1.

### 3.3. Drug-like Filter

Criteria from the “blockbuster” filter available in the *FILTER* module of the *OMEGA* v.4.1.2.0 software (OpenEye Scientific Software, Santa Fe, NM, USA) [47,48] were used to filter compounds selected through shape-based screening for potentially “drug-like” compounds. The applied criteria are based on physicochemical properties (e.g., molecular weight, solubility, clogP, and number of hydrogen bond donors/acceptors), topological properties (e.g., number of rotatable and rigid bonds, chiral centers, and ring systems), atomic and chemical group contents (e.g., number of carbon atoms, “heteroatoms”, “undesirable” chemical groups, such as protein-reactive electrophilic groups like acyl halides, aldehydes, epoxides, and Michael acceptors; redox cyclers, like quinones; and metal chelators). “Secondary filters” based on empirical rules, such as “Lipinski’s Rule of Five” (modified to allow a maximum of three violations of the established criteria), as well as filters for known “aggregators”, were also included as part of the “blockbuster” criteria.

### 3.4. Protein Structure Preparation for Docking Procedures

The crystal structure of the P2X7 receptor from *Ailuropoda melanoleuca* (*Am*P2X7) was chosen according to its high sequence identity (~85%) to human P2X7. This 3D crystal structure, complexed with the allosteric antagonist JNJ-47965567 [46], was obtained from the Protein Data Bank in PDB format (PDB ID: 5U1X). The biological assembly (trimeric structure) of *Am*P2X7 was built using the *PyMOL* v.2.5.0 software (Schrödinger, New York, NY, USA), based on the alignment of *Am*P2X7 monomers with the crystal structure of P2X7 from *Rattus norvegicus*, available in PDB in the trimeric form (PDB ID: 6U9V, resolution: 2.90 Å) [52]. Then, the protein structure was prepared for docking procedures using the *MAESTRO* v.13.5 software (Schrödinger, New York, NY, USA). All water molecules and ligands (including the JNJ-47965567 antagonist and the NAG molecules) were removed from the original PDB file. All hydrogen atoms were added and the protonation states of the ionizable residues at pH 7.4 were assigned.

### 3.5. Docking Procedures and Visual Inspection

Docking calculations were performed using the *GOLD* v.5.2 (CCDC, Cambridge, UK) [53] and *DockThor* v.2.0 (LNCC, Petrópolis, Brazil) programs, with the *Am*P2X7 structure prepared as described above. For docking using the *GOLD* program, the protein binding site region was defined considering all atoms within a 10 Å radius from the JNJ-47965567 antagonist as observed in the *Am*P2X7-JNJ-47965567 complex (PDB ID: 5U1X) [46]. Ten docking runs were performed for each compound using the ChemPLP scoring function [54] and default settings for genetic algorithm parameters. Docking procedures using the *DockThor* program were conducted using the following grid center coordinates, x (171.206 Å), y (157.196 Å), and z (223.025 Å), and grid size, x (20 Å), y (20 Å), and z (20 Å), as well as default virtual screening parameters for soft docking. Twelve docking runs were performed for each compound using the MMFF94S force field as a scoring function [55].

After performing the docking runs, the compounds were ranked according to the score values obtained in each program independently. The 25 compounds with the highest score values from the MEGx database and the 100 compounds with the highest score values from the NATx database were then visually inspected. The best scoring pose of each docked compound was analyzed by visual inspection into the *Am*P2X7 binding site using the *PyMOL* v.2.5.0 (Schrödinger, New York, NY, USA), *LigandScout* v.4.4 (Inte:Ligand, Viena, Austria) [56], and PLIP v.2.2.0 (Biotechnology Center TU Dresden, Dresden, Germany) [57] programs. Next, a consensus analysis was performed to detect identical molecules in the top-25 or top-100 rankings of the two docking programs.

Docking procedures using *GOLD* and *DockThor* programs were validated by redocking JNJ-47965567 and four other *Am*P2X7 antagonists co-crystallized with *Am*P2X7 (PDB ID: 5U1U, 5U1V, 5U1W, and 5U1Y; resolution values ranging from 3.30 Å to 3.60 Å) into their corresponding allosteric binding sites in the *Am*P2X7 structure, using the same settings as defined above [46].

### 3.6. Molecular Dynamics Simulations

#### 3.6.1. Ligand Parameterization

The protonation state and ligand charges were set using the *Chimera* (UCSF, San Francisco, CA, USA) at pH 7. The ligand parameter files required for carrying out the MD steps were generated using the Antechamber program [58], employing a set of scripts known as *ACPYPE* [59]. The Lennard–Jones molecular parameters and bonded interactions were obtained using the General Amber Force Field (GAFF) [60]. Atom partial charges were assigned using the *AM1-BCC* generator [61].

#### 3.6.2. Geometric Configuration of Simulated Systems 

An orthorhombic box with approximate dimensions of 11 × 11 × 18 nm (in X, Y, and Z directions, respectively) was constructed under periodic boundary conditions and explicit solvation with 56,085 Transferable Intermolecular Potential with 3-points (TIP3P) water molecules [62]. System neutrality was achieved by adding 110 Na^+^ ions and 134 Cl^−^ ions at physiological concentration.

All protein parameterizations were conducted using *Gromacs* v.2023.3 software [63]. The Amber99SB.ff force field [64] was applied for all systems. Equilibration was performed at a pressure of 1.013 Bar and a temperature of 310 K in the Gibbs ensemble (NPT), utilizing a 2 fs integration time step. The non-bonded interactions’ cutoff (Lennard–Jones 6-12 potential) and treatment of the Particle-mesh Ewald (PME) electrostatic potential [65] were set using algorithmic automation (*Verlet*) [66]. The *LINCS* algorithm was employed for covalent bonds [67,68].

The energy minimization structural optimization process was conducted in two stages: initially, using the steepest descent algorithm with 20,000 minimization steps; subsequently, an additional 20,000 steps using the conjugate gradient algorithm, with a gradient tolerance of <1.0 kJ mol^−1^. Throughout equilibration, system heating commenced at different temperatures per replica (0.5, 0.6, and 0.7 K, respectively), and atom velocities were generated using the Maxwell–Boltzmann distribution for each initial heating temperature, gradually increasing by 1 K after every 50 ps of simulation until reaching the final reference temperature of 310 K. Positions of protein and ligand heavy atoms were restrained using a harmonic potential with force constants, decreasing after every 3 ns of simulation at values of 4000, 2000, 1000, 750, 500, and finally 250 kJ/mol^−1^ nm^−2^. Following energy minimization, thermalization, and equilibration steps, production dynamics were executed to obtain atom trajectories, employing the NPT ensemble with a Parrinello–Rahman thermostat and Nose–Hoover barostat at a constant temperature of 310 K and pressure of 1.013 Bar. 

#### 3.6.3. Analysis of Trajectories Simulated by Molecular Dynamics 

The root-mean-square deviation (RMSD) was calculated to assess the deviation of a structure from its reference conformation over the simulation time. The RMSD value was computed using the *gmx rms* program through the following expression: $$RMSD(t_{1},t_{2})={\left[\frac{1}{M}\sum_{i=1}^{N}mi{\left\|r_{i}\left(t_{1}\right)-r_{i}(t_{2})\right\|}^{2}\right]}^{\frac{1}{2}}$$
where *t*_1_ is an instant in time *t*, *t*_2_ correspond to the structure at the reference instant, $M=\sum_{i=1}^{n}\;mi$, *r_i_* (*t*) represents the position of atom *i* at time *t*, and *N* is the number of atoms in the system.

## 4. Conclusions

The VS approach reported herein, comprising shape-based screening, drug-like filtering, and docking followed by careful visual inspection into an allosteric P2X7 binding site, was proven useful in screening two natural product databases (MEGx and NATx) to search potential P2X7 antagonists. Using this approach, four compounds from the MEGx database and four from the NATx database were selected as potential P2X7 antagonists, representing a >99% reduction in the number of compounds from each database. To the best of our knowledge, this study is the first to combine shape-based screening (a ligand-based approach) and docking followed by careful visual inspection (a structure-based approach) to search for novel potential P2X7 antagonists from natural product-derived compound databases. Similar approaches could be useful for selecting inhibitors/antagonists of other receptors and/or biological targets. The compounds selected herein represent structurally novel and drug-like P2X7 antagonist candidates. This study, therefore, allows for the exploration of natural product-derived compounds as novel P2X7 antagonist candidates. 

## Figures and Tables

**Figure 1 pharmaceuticals-17-00592-f001:**
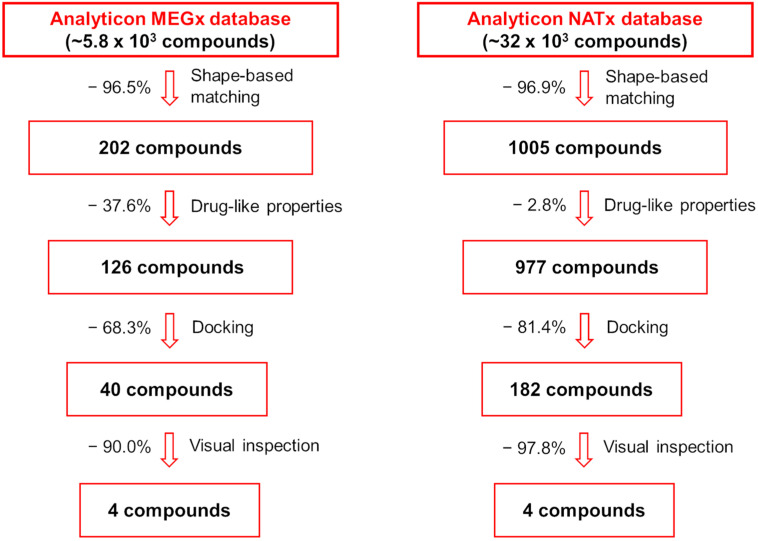
Schematic representation of the hierarchical sequence of selection filters applied to the MEGx and NATx databases aiming at the selection of potential P2X7 antagonists.

**Figure 2 pharmaceuticals-17-00592-f002:**
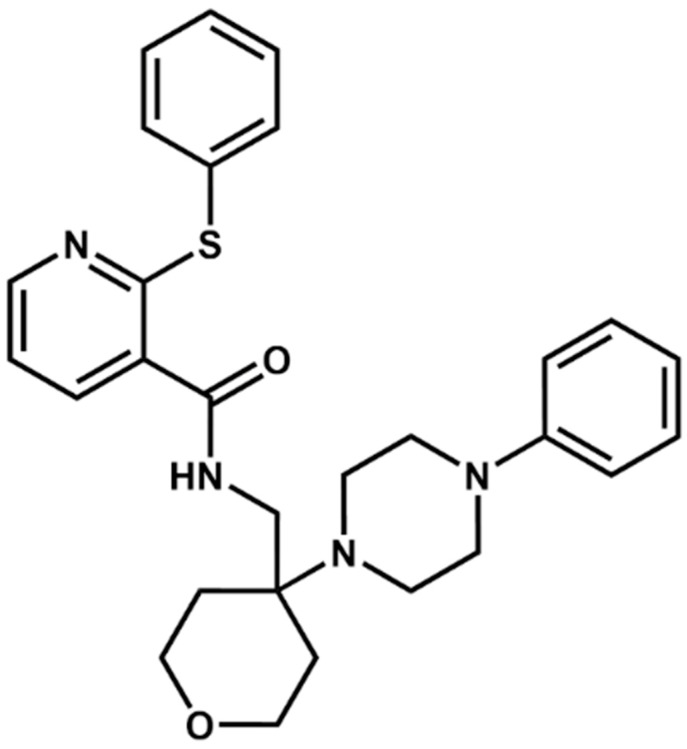
Two-dimensional representation of the structure of the JNJ-47965567, a selective P2X7 antagonist.

**Figure 3 pharmaceuticals-17-00592-f003:**
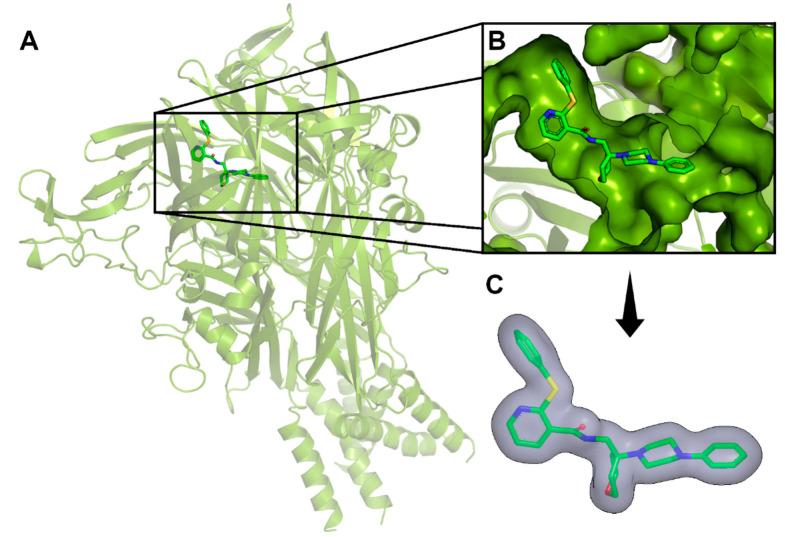
Shape-based model building. (**A**) Schematic representation of the *Am*P2X7-JNJ-47965567 complex (PDB ID: 5U1X; resolution: 3.20 Å). (**B**) Close-up view of JNJ-47965567 bound to the allosteric *Am*P2X7 binding site. (**C**) The shape-based model was generated using *ROCS*. The molecular shape surface is represented in gray. JNJ-47965567 is represented as sticks, with carbon, oxygen, nitrogen, and sulfur atoms colored in green, red, blue, and yellow, respectively. The figure was prepared using *PyMOL* v.2.5.0 and *ROCS* v.3.5.0.2.

**Figure 4 pharmaceuticals-17-00592-f004:**
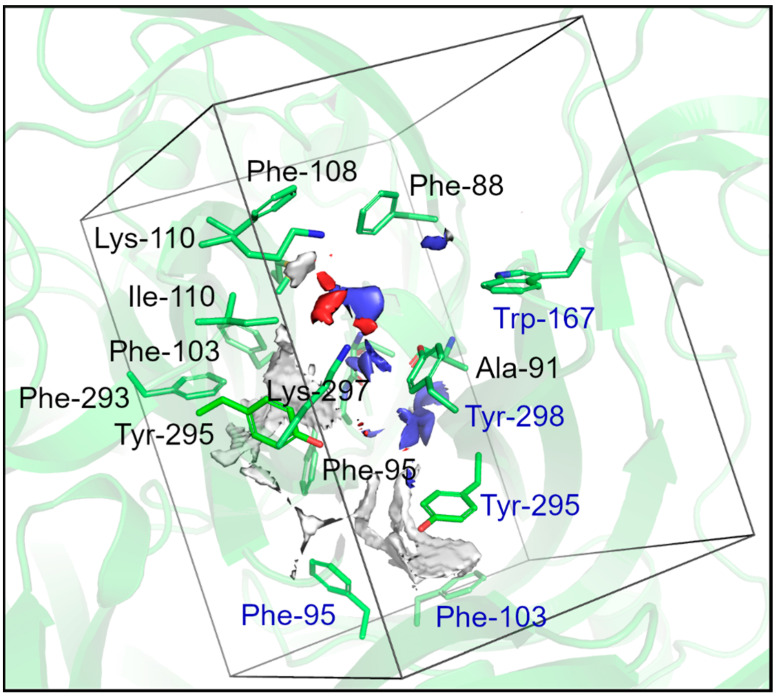
Representation of the molecular interaction fields (MIFs) calculated using the GRID program for the allosteric binding site of *Am*P2X7 used as the reference pocket for the docking calculations in our VS protocol. White, blue, and red surfaces represent the hydrophobic, hydrogen-bond donor, and hydrogen-bond acceptor MIFs obtained using the DRY probe (energy cutoff value of −1.0 kcal.mol^−1^), N1 probe (energy cutoff value of −7.0 kcal.mol^−1^), and O probe (energy cutoff value of −7.0 kcal.mol^−1^), respectively. The main binding site residues are represented as sticks, and the protein 3D structure is shown as a cartoon representation. Residues from subunits A and B are labeled in black and blue, respectively. The figure was prepared using *PyMOL* v.2.5.0.

**Figure 5 pharmaceuticals-17-00592-f005:**
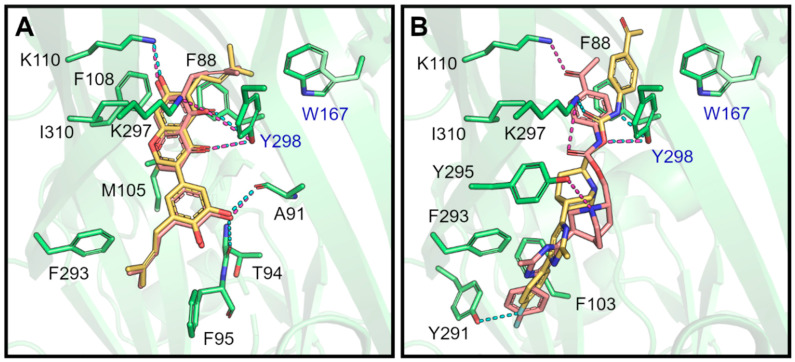
Schematic representation of the predicted binding modes (docking solutions with the best score) of two representative compounds selected by our VS protocol as potential P2X7 antagonists. (**A**) Superposition between the best scoring poses from the *GOLD* (carbon atoms in yellow) and *DockThor* (carbon atoms in pink) programs for compound NP-016468 from the MEGx database. (**B**) Superposition between the best scoring poses from the *GOLD* (carbon atoms in yellow) and *DockThor* (carbon atoms in pink) programs for compound NAT13-340161 from the NATx database. The *Am*P2X7 structure is represented as a green cartoon. Representative compound structures and binding site residues that make hydrophobic and/or hydrogen bond interactions with the representative compounds are shown as sticks. Oxygen, nitrogen, and fluorine atoms are colored red, blue, and light blue, respectively. Hydrogen bonds are represented by dashed lines (in cyan for *GOLD* docking poses and in magenta for *DockThor* docking poses). Residues from different subunits are labeled in black and in blue. The figure was prepared using the *PyMOL* v.2.5.0 software.

**Figure 6 pharmaceuticals-17-00592-f006:**
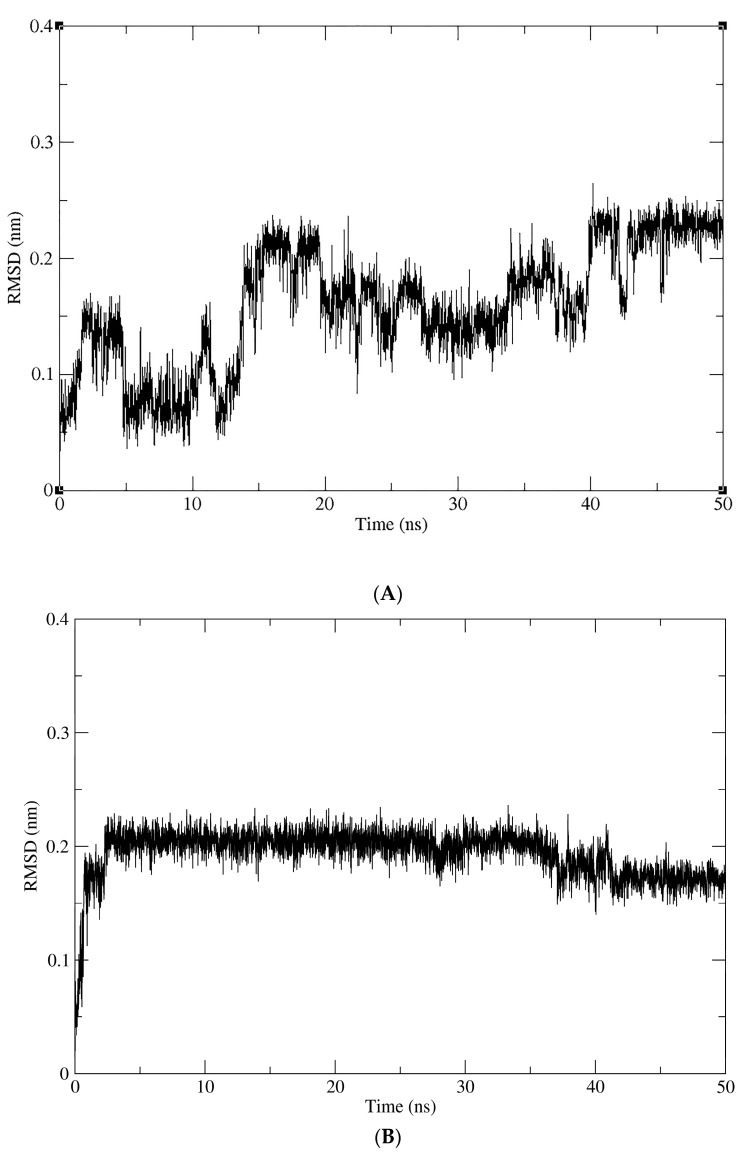
Variation in the RMSD of the compounds complexed with the P2X7 receptor over the simulation time. (**A**) NP-016468. (**B**) NP-025357. (**C**) NP-025047.

**Figure 7 pharmaceuticals-17-00592-f007:**
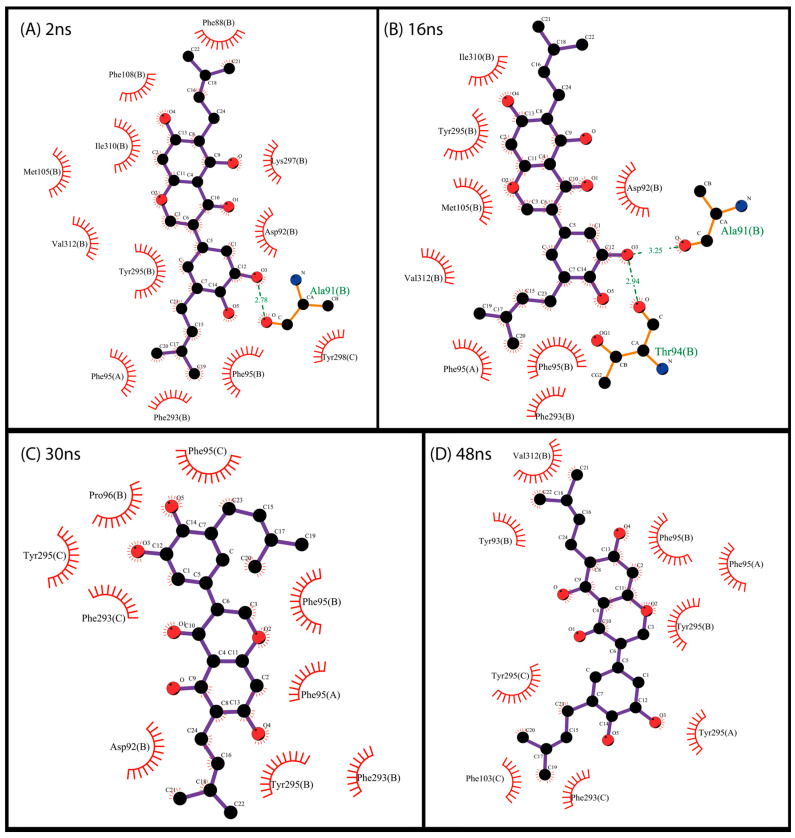
Interaction map between protein residues and compound NP-016468 using the most representative structures obtained over 50 ns, clustered using the gromos method (cut-off 0.14 Å). Residues are identified by their names, position in the primary structure, and chain. Each of the most representative structures is associated with an average frame of the computational simulation. The maps were generated using the free version of the LigPlot+ v.2.2 program.

**Figure 8 pharmaceuticals-17-00592-f008:**
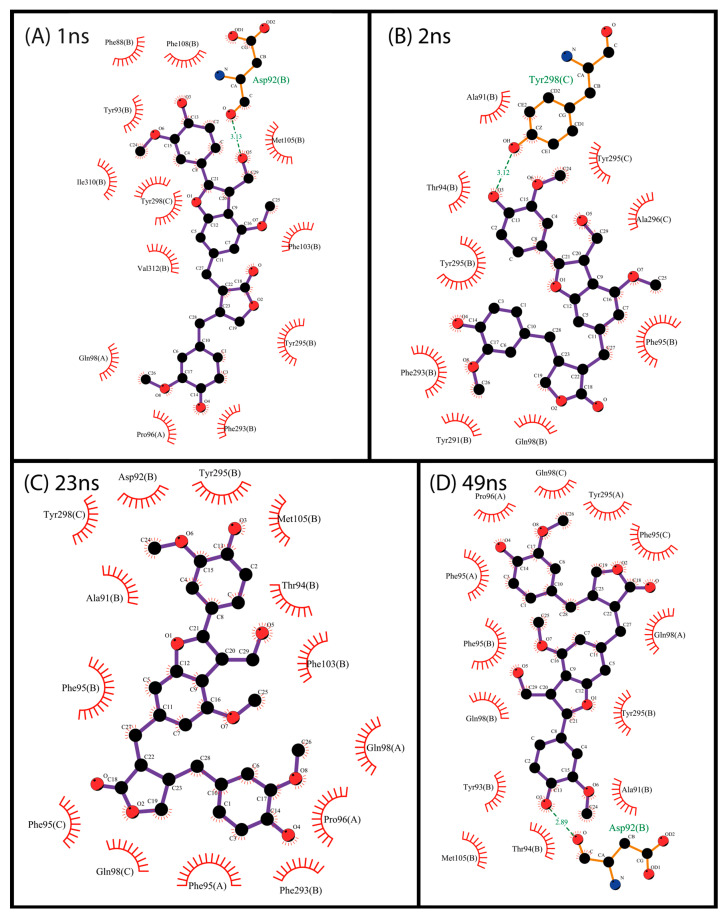
Interaction map between protein residues and compound NP-025357 using the most representative structures obtained over 50 ns, clustered using the gromos method (cut-off 0.14 Å). Residues are identified by their names, position in the primary structure, and chain. Each of the most representative structures is associated with an average frame of the computational simulation. The maps were generated using the free version of the LigPlot+ v.2.2 program.

**Figure 9 pharmaceuticals-17-00592-f009:**
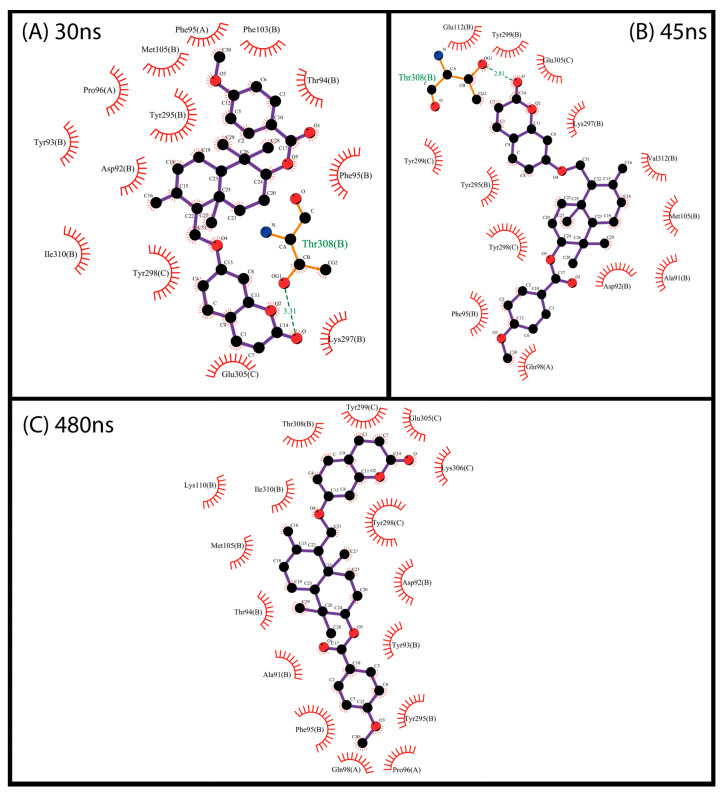
Interaction map between protein residues and compound NP-025047 using the most representative structures obtained over 50 ns, clustered using the gromos method (cut-off 0.14 Å). Residues are identified by their names, position in the primary structure, and chain. Each of the most representative structures is associated with an average frame of the computational simulation. The maps were generated using the free version of the LigPlot+ v.2.2 program.

**Table 1 pharmaceuticals-17-00592-t001:** Some physicochemical properties * and docking scores of the selected compounds from the MegX and NatX databases as potential P2X7 antagonists.

DB	Compound (Structure)	Molecular Formula	MW (g∙mol^−1^)	clogP	TPSA (Å)	HA	HD	*GOLD ChemPLP* Score	*DockThor* Score
**MegX**	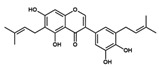 NP-016468	C_25_H_26_O_6_	422.47	6.23	107.22	6	4	89.99	−10.924
	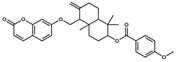 NP-025047	C_32_H_36_O_6_	516.63	6.67	71.06	4	0	94.67	−12.169
	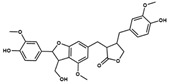 NP-025357	C_30_H_32_O_9_	536.57	3.77	123.91	8	3	101.02	−11.800
	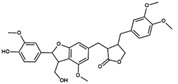 NP-025358	C_31_H_34_O_9_	550.60	3.91	112.91	8	2	99.42	−11.984
**NatX**	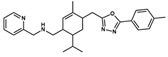 NAT28-412055	C_27_H_34_N_4_O	430.59	4.66	63.84	4	1	99.19	−11.061
	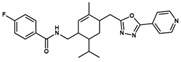 NAT28-416626	C_26_H_29_FN_4_O_2_	448.54	3.72	80.91	4	1	101.31	−11.352
	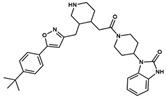 NAT14-350419	C_33_H_41_N_5_O_3_	555.72	4.03	90.71	4	2	99.97	−11.554
	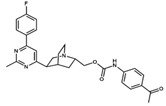 NAT13-340161	C_28_H_29_FN_4_O_3_	488.56	4.62	84.42	6	1	99.96	−11.028

DB: database; MW: molecular weight; TPSA: topological polar surface area; HA: hydrogen acceptors; HD: hydrogen donors. * Taken from AnalytiCon MegX and NatX databases.

**Table 2 pharmaceuticals-17-00592-t002:** Protein–ligand interactions recognized between the best scoring docking poses of the compounds selected from the MEGx database as potential P2X7 antagonists.

	Compound							
	NP-016468		NP-025047		NP-025357		NP-025358	
Residues Subunit A	*GOLD*	*DockThor*	*GOLD*	*DockThor*	*GOLD*	*DockThor*	*GOLD*	*DockThor*
F88	HD	HD	HD	HD	HD		HD	
A91	HB	HB						
D92								HD
Y93								HB
T94	HB		HB					
F95	HD	HD	HD	HD	HD	HD	HD	HD
P96				HD				
F103				HD	HD		HD	
M105	HD		HD	HD				
F108	HD							
K110	HB	HB				HB	HB	HB
F293	HD	HD		HD	HD		HD	
Y295		HD	HD		HB	HB, π-π	HB, HB	HD, HB, π-π
K297		HB	HB	π-C, SB		HB		π-π
I310	HD	HD	HD	HD		HD		HD
V312			HD	HD		HD		
Residues Subunit B	*GOLD*	*DockThor*	*GOLD*	*DockThor*	*GOLD*	*DockThor*	*GOLD*	*DockThor*
W167	HD				HD		HD	
F293				HD				
Y295			HD	HD				
A296				HB				
Y298	HB	HD, HB	HD		HD, HB		HD	HD
Residues Subunit C	*GOLD*	*DockThor*	*GOLD*	*DockThor*	*GOLD*	*DockThor*	*GOLD*	*DockThor*
F95	HD							

HD: hydrophobic interaction; HB: hydrogen bond interaction; SB: salt bridge; π-π: π-stacking interaction. Interactions were recognized using the *LigandScout* and *PLIP* programs.

**Table 3 pharmaceuticals-17-00592-t003:** Protein–ligand interactions recognized between the best scoring docking poses of the compounds selected from the NATx database as potential P2X7 antagonists.

	Compound ID							
	NAT28-412055		NAT28-416626		NAT14-350419		NAT13-340161	
Residues Subunit A	*GOLD*	*DockThor*	*GOLD*	*DockThor*	*GOLD*	*DockThor*	*GOLD*	*DockThor*
F88	HD	HD	HD		HD	HD	HD	HD
A91								
D92		HD						HD
T94								
F95	HD	HD	HD	HD	HD	HD	HD	HD
F103		HD	HD	HD		HD	HD	HD
M105	HD		HD		HD			
F108				π-π		HD		
K110				HB		HD		HB
Y291							HB	
F293	HD	HD	HD	HD		HD	HD	HD
Y295	HD		HD	π-π	HB	HD	HD	HB
K297	HBA		HBA	π-C	π -C	π-C	HB	HB
I310	HD	HD	HD	HD	HD	HD		HD
V312	HD		HD					
Residues Subunit B	*GOLD*	*DockThor*	*GOLD*	*DockThor*	*GOLD*	*DockThor*	*GOLD*	*DockThor*
F95		HD			HD			
W167			HD		HD		HD	
Y293		HD						HD
Y295	HD				HD	HD		
A296	HD		HD					
Y298	HD	HD		HD	HD	HD, HB	HB	HB
Residues Subunit C	*GOLD*	*DockThor*	*GOLD*	*DockThor*	*GOLD*	*DockThor*	*GOLD*	*DockThor*
F95	HD							
P96				HD		HD		HD
Q98			HB				HB	

HD: hydrophobic interaction; HB: hydrogen bond interaction; π-π: π-stacking interaction; π-C: π-cation interaction. Interactions were recognized using the *LigandScout* and *PLIP* programs.

## Data Availability

Data are contained within the article and the Appendix A.

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
