# Peer review of "A Hybrid Approach Combining Shape-Based and Docking Methods to Identify Novel Potential P2X7 Antagonists from Natural Product Databases"

_pharmaceuticals, 2024, doi:10.3390/ph17050592_

Round 1

Reviewer 1 Report (Previous Reviewer 1)

Comments and Suggestions for Authors

The manuscript has undergone significant improvements, however, I regret to recommend rejection as I have yet to see any wet lab validation.

Author Response

Dear Reviewer,

We appreciate all comments and suggestions on the manuscript entitled “A hybrid approach combining shape-based and docking methods to identify novel potential P2X7 antagonists from natural product database". We have considered all comments, as displayed below in a point-by-point response to each issue and/or comment. The text modifications are highlighted in the text. We hope that the manuscript is now publishable in the journal. 

Comments and Suggestions for Authors

The manuscript has undergone significant improvements, however, I regret to recommend rejection as I have yet to see any wet lab validation.”

Response: Thank you for your comment regarding the manuscript improvements. In vitro validation will be conducted soon. Unfortunately, it was not possible to carry it out due to the deadline for this special issue. The limitation of these assays is that the molecules need to be synthesized, which can take a few months. Furthermore, the molecules need to be imported into Brazil, which is a bureaucratic process due to Brazilian laws, which significantly increases the acquisition time and impacts our research. To improve analyses beyond docking, we performed molecular dynamics tests to monitor the interaction of ligands at the P2X7 receptor binding site. Despite this, in the revised version of the manuscript, 79 known P2X7 allosteric antagonists (from the literature) were applied to our generated shape-based model to validate our proposed shape-based approach.

Reviewer 2 Report (Previous Reviewer 3)

Comments and Suggestions for Authors

The authors have edited the paper appropriately since the original submission, and this is a great manuscript.

Author Response

Dear Reviewer,

We appreciate all comments and suggestions on the manuscript entitled “A hybrid approach combining shape-based and docking methods to identify novel potential P2X7 antagonists from natural product database". We have considered all comments, as displayed below in a point-by-point response to each issue and/or comment. The text modifications are highlighted in the text. We hope that the manuscript is now publishable in the journal. 

Comments and Suggestions for Authors

The authors have edited the paper appropriately since the original submission, and this is a great manuscript.”

Response: Thank you for your positive comment.

Reviewer 3 Report (Previous Reviewer 2)

Comments and Suggestions for Authors

Authors have improved the manuscript and it can be accepted but still there is no active wet lab based study to prove the hypothesis.

Author Response

Dear Reviewer,

We appreciate all comments and suggestions on the manuscript entitled “A hybrid approach combining shape-based and docking methods to identify novel potential P2X7 antagonists from natural product database". We have considered all comments, as displayed below in a point-by-point response to each issue and/or comment. The text modifications are highlighted in the text. We hope that the manuscript is now publishable in the journal. 

Comments and Suggestions for Authors

 Authors have improved the manuscript and it can be accepted but still there is no active wet lab based study to prove the hypothesis.”

Response: Thank you for your comment regarding the manuscript improvements. In vitro validation will be conducted soon. Unfortunately, it was not possible to carry it out due to the deadline for this special issue. The limitation of these assays is that the molecules need to be synthesized, which can take a few months. Furthermore, the molecules need to be imported into Brazil, which is a bureaucratic process due to Brazilian laws, which significantly increases the acquisition time and impacts our research. To improve analyses beyond docking, we performed molecular dynamics tests to monitor the interaction of ligands at the P2X7 receptor binding site. Despite this, in the revised version of the manuscript, 79 known P2X7 allosteric antagonists (from the literature) were applied to our generated shape-based model to validate our proposed shape-based approach.

Reviewer 4 Report (New Reviewer)

Comments and Suggestions for Authors

In the manuscript titled “A hybrid approach combining shape-based and docking methods to identify novel potential P2X7 antagonists from natural products databases”, the authors report a “classical” computational drug discovery study, in order to discover potential molecules as P2X7 antagonists, starting from two databases containing around 5.8x103 and 3.2x104 compounds respectively. From these databases after a shape-based matching, drug-like properties and docking were selected four compounds from each database. The topic of this study is interesting, but the manuscript has issues.

My comments are:

1. The introduction needs to be improved.

2. The authors have used as protein target the low resolution (3.20 Å) X-ray structure of Ailuropoda melanoleuca P2X7, which is different from the human one. How the authors have established, that the “allosteric binding pocket” of Ailuropoda melanoleuca have the same amino acid sequence than the one of human protein? The authors should report, not only the values of global similarity (85.8%), but the similarity or equality of amino acids types in allosteric site, which are involved in interactions with the ligands. For this purpose, the authors should perform and show the two compared sequences (from Homo sapiens and from Ailuropoda melanoleuca) of P2X7 by a specific server (e.g. Protein Blast, TM-allign, Sequence Manipulation Suite or others) or by other software (see Fiser, Template-Based Protein Structure Modeling, Methods Mol Biol. 2010; 673: 73–94. DOI: 10.1007/978-1-60761-842-3_6). All these information need to be reported in supplementary. If the sequence in the allosteric site is not adequate, the authors have should to obtain a homology model of human protein using fasta human sequence and modelling it from PDB-ID:5U1X by using, for example, the online server SWISS-MODEL or similar.

3.It is not reported a table containing both of GOLD and DockThor docking score values for the most promising compounds, in order to compare them with the docking score values of original X-ray structures reported in supplementary. Please provide to add at Table 1 two columns reporting the docking scores.

4. Due to the fact that molecular docking gives a “static” vision of interactions (due to the protein is treated as a rigid body), in order to evaluate in a more realistic way the interaction energy and ligand-P2X7 stability of the most promising molecules (reported in Table 1), the authors should perform on them the molecular dynamics simulation.

In conclusion, the manuscript , although presents a lot of data,  doesn't give sufficient information to assert that the selected molecules could be biologically active on human P2X7.  It needs a deep revision following as required at points 1-4.

Author Response

Dear Reviewer,

We appreciate all comments and suggestions on the manuscript entitled “A hybrid approach combining shape-based and docking methods to identify novel potential P2X7 antagonists from natural product database". We have considered all comments, as displayed below in a point-by-point response to each issue and/or comment. The text modifications are highlighted in the text. We hope that the manuscript is now publishable in the journal. 

Comments and Suggestions for Authors

In the manuscript titled “A hybrid approach combining shape-based and docking methods to identify novel potential P2X7 antagonists from natural products databases”, the authors report a “classical” computational drug discovery study, in order to discover potential molecules as P2X7 antagonists, starting from two databases containing around 5.8x103 and 3.2x104 compounds respectively. From these databases after a shape-based matching, drug-like properties and docking were selected four compounds from each database. The topic of this study is interesting, but the manuscript has issues.

My comments are:

  1. The introduction needs to be improved.

Response: Thank you for your comment. To improve the Introduction section, we re-added some sentences about the limitations of the docking approach, which were in the original version of the manuscript.

  1. The authors have used as protein target the low resolution (3.20 Å) X-ray structure of Ailuropoda melanoleucaP2X7, which is different from the human one. How the authors have established, that the “allosteric binding pocket” of Ailuropoda melanoleuca have the same amino acid sequence than the one of human protein? The authors should report, not only the values of global similarity (85.8%), but the similarity or equality of amino acids types in allosteric site, which are involved in interactions with the ligands. For this purpose, the authors should perform and show the two compared sequences (from Homo sapiens and from Ailuropoda melanoleuca) of P2X7 by a specific server (e.g. Protein Blast, TM-allign, Sequence Manipulation Suite or others) or by other software (see Fiser, Template-Based Protein Structure Modeling, Methods Mol Biol. 2010; 673: 73–94. DOI: 10.1007/978-1-60761-842-3_6). All these information need to be reported in supplementary. If the sequence in the allosteric site is not adequate, the authors have should to obtain a homology model of human protein using fasta human sequence and modelling it from PDB-ID:5U1X by using, for example, the online server SWISS-MODEL or similar.

Response: Thanks for your comment. We clarify that all residues that form the allosteric binding site are conserved among Ailuropoda melanoleuca P2X7 and the Homo sapiens P2X7 structures, as revealed by the alignment of the amino acid sequences of P2X7 from these two species using BLAST. This was included in the revised version of the manuscript in the Results and Discussion section:

“(…) Notably, AmP2X7 has a high sequence identity (85.8%) to the Homo sapiens P2X7 (HsP2X7), and all the residues that form the allosteric binding site are conserved among AmP2X7 and HsP2X7 (Supplementary Material). (…)”

  1. It is not reported a table containing both of GOLD and DockThor docking score values for the most promising compounds, in order to compare them with the docking score values of original X-ray structures reported in supplementary. Please provide to add at Table 1 two columns reporting the docking scores.

Response: Thanks for your suggestion. In the revised version of the manuscript, we added two columns to Table 1, showing the GOLD and DockThor docking score values for the compounds selected by our VS approach. We also added a brief discussion about that in the Results and Discussion section.

“(…) The structures of selected compounds, some of their physicochemical properties, and their GOLD ChemPLP and DockThor score values are shown in Table 1. Notably, the GOLD ChemPLP score values for the compounds selected by our VS protocol (89.99 to 101.31) are overall higher than the score values of the five known AmP2X7 antagonists aforementioned (67.35 to 97.22), which were redocked into the AmP2X7 allosteric binding site (Supplementary Material). Additionally, the DockThor score values for the compounds selected by VS (-12.169 to -10.924) are lower (more negative) than the score values of the five known AmP2X7 antagonists (-11.242 to -9.601; Supplementary Material). These results suggest that the compounds selected by our VS would have higher affinities to AmP2X7 in comparison to the known AmP2X7 antagonists. (…)”

  1. Due to the fact that molecular docking gives a “static” vision of interactions (due to the protein is treated as a rigid body), in order to evaluate in a more realistic way the interaction energy and ligand-P2X7 stability of the most promising molecules (reported in Table 1), the authors should perform on them the molecular dynamics simulation.

Response: Thank you for highlighting this important aspect. Acknowledging the significance of protein flexibility in understanding protein-ligand interactions, we conducted Molecular Dynamics (MD) experiments to evaluate this dynamic behavior. It is important to note that performing comprehensive MD analyses for all molecules involves considerable time investment, particularly for extensive simulations lasting around 500 ns and having more than 200k atoms, as recommended for a detailed study.

Regrettably, due to time constraints imposed by the journal's deadline, which allowed only a 30-day extension, we were unable to conduct prolonged MD simulations for all the reported molecules. However, we performed MD simulations for select promising molecules to assess their stability with P2X7, aiming to capture the dynamic aspects of these interactions within the available timeframe.

We acknowledge the importance of a more comprehensive analysis and intend to further explore and expand our MD simulations in future studies, allowing for a more comprehensive understanding of protein-ligand dynamics in this context.

 In conclusion, the manuscript , although presents a lot of data,  doesn't give sufficient information to assert that the selected molecules could be biologically active on human P2X7.  It needs a deep revision following as required at points 1-4.

Response: Thank you for your comment regarding the manuscript improvements. In vitro validation will be conducted soon. Unfortunately, it was not possible to carry it out due to the deadline for this special issue. The limitation of these assays is that the molecules need to be synthesized, which can take a few months. Furthermore, the molecules need to be imported into Brazil, which is a bureaucratic process due to Brazilian laws, which significantly increases the acquisition time and impacts our research. To improve analyses beyond docking, we performed molecular dynamics tests to monitor the interaction of ligands at the P2X7 receptor binding site. Despite this, in the revised version of the manuscript, 79 known P2X7 allosteric antagonists (from the literature) were applied to our generated shape-based model to validate our proposed shape-based approach.

Round 2

Reviewer 1 Report (Previous Reviewer 1)

Comments and Suggestions for Authors

The authors have made a great effort to improve the manuscript.

Reviewer 4 Report (New Reviewer)

Comments and Suggestions for Authors

The authors have improved the manuscript, and it can be accepted for publication.

This manuscript is a resubmission of an earlier submission. The following is a list of the peer review reports and author responses from that submission.

Round 1

Reviewer 1 Report

Comments and Suggestions for Authors

The manuscript by Alves et al described a virtual screening approach to identify new P2X7 receptor antagonists from natural product databases, potentially contributing to the development of therapies for inflammation. These compounds showed structural differences from known antagonists, exhibited drug-like properties, and were predicted to interact with key residues in the allosteric binding pocket.

The lack of functional experiments is a limitation of this study, which introduces uncertainty to the computational results. While virtual screening and docking approaches can provide valuable insights, experimental validation should always be followed to confirm the prediction.

There are numerous P2X7 receptor inhibitors, many of which have failed in clinical trials. How do the newly identified compounds using the methods described in this paper overcome some of the shortcomings of previous compounds?

Although the JNJ was co-crystallized with AmP2X7, shape-based screening can use the structure of AmP2X7. However, for docking, it is recommended to use the structure of rat P2X7.

Page 2, line 53. The direction of ion flow is not only determined by the electrochemical gradient difference but also by the concentration difference.

Author Response

Dear Reviewer,

We appreciate all comments and suggestions on the manuscript entitled “Combining shape-based and docking approaches for identifying potential P2X7 antagonist candidates from natural product libraries". We have considered all comments, as displayed below in a point-by-point response to each issue and/or comment. The text modifications are highlighted in the text. We hope that the manuscript is now publishable in the journal. 

Comments and Suggestions for Authors

The manuscript by Alves et al described a virtual screening approach to identify new P2X7 receptor antagonists from natural product databases, potentially contributing to the development of therapies for inflammation. These compounds showed structural differences from known antagonists, exhibited drug-like properties, and were predicted to interact with key residues in the allosteric binding pocket.

POINT 1: The lack of functional experiments is a limitation of this study, which introduces uncertainty to the computational results. While virtual screening and docking approaches can provide valuable insights, experimental validation should always be followed to confirm the prediction.

Response 1: Thank you for your comment. We completely agree that experimental validation is extremely important for confirming the predicted biological activities of compounds selected by virtual screening. We additionally agree that the lack of experimental validation is a limitation of our study. This point was discussed in several parts of our manuscript, as follows:

Abstract (last sentence, page 2, Revised Manuscript; also present in the Original Manuscript):

“(…) Therefore, the combination of shape-based screening and docking approaches applied in our study is proven useful in selecting potential novel P2X7 antagonist candidates from natural product-derived compound databases, which should be submitted to in vitro assays for the experimental validation of our VS.”

Introduction section (4th paragraph, page 3, Revised Manuscript; also present in the Original Manuscript):

“(…) Compounds selected by VS campaigns must, nevertheless, be submitted to confirmatory in vitro and/or in vivo assays for VS experimental validation [26,27,29–31].”

Introduction section (6th paragraph, page 4, Revised Manuscript; also present in the Original Manuscript):

“(…) In this regard, experimental validation steps are extremely relevant to overcome inherent docking limitations [31].”

Introduction section (last paragraph, page 4, Revised Manuscript; also present in the Original Manuscript):

“(…) The selected compounds represent structurally novel drug-like natural product-derived P2X7 antagonist candidates, which should be tested in in vitro assays for the experimental validation of our VS approach.”

Conclusions section (page 18, Revised Manuscript; also present in the Original Manuscript):

“(…) The compounds selected herein represent structurally novel and drug-like P2X7 antagonist candidates, which should be tested in in vitro assays for the experimental validation of our VS approach. (…)”

Additionally, in the revised version of the manuscript, we added the following sentences in the last paragraph of the Results and Discussion section, to discuss the importance of the experimental validation of VS models:

Results and Discussion section (last paragraph, page 16, Revised Manuscript):

“(…) Finally, we point out that these compounds should be tested in in vitro assays for the experimental validation of our VS. As mentioned in the Introduction section, the experimental validation is extremely important to confirm the utility of a VS approach due to the inherent limitations of in silico methods.”

POINT 2: There are numerous P2X7 receptor inhibitors, many of which have failed in clinical trials. How do the newly identified compounds using the methods described in this paper overcome some of the shortcomings of previous compounds?

Response 2: Despite the limitation regarding in vitro assays, our intention with this work was to propose a new virtual screening model applied to the P2X7 receptor, since studies using this methodology are still quite scarce and very recent (doi: 10.15586/ijfs.v35i2.2288, 10.1111/cbdd.13867, 10.1016/j.compbiomed.2023.107299,10.3389/fphar.2022.1094607, 10.1016/j.bcp.2016.07.020). In this way, candidate compounds were identified based on the shape of the JNJ-47965567 antagonist, also presenting drug-like properties and, according to predictions, they are capable of interacting with key residues of the allosteric site of the P2X7 receptor. Therefore, the innovation that this methodology brings is to provide more robustness than conventional docking tests used in virtual screening.

POINT 3: Although the JNJ was co-crystallized with AmP2X7, shape-based screening can use the structure of AmP2X7. However, for docking, it is recommended to use the structure of rat P2X7.

Response 3: Thank you for your valuable suggestion. However, we opted to use the panda P2X7 receptor due to its higher homology to humans than rat species, i.e., ~85% (doi: 10.7554/eLife.31186) compared to 80% of rats (doi: 10.1007/s11302-023-09957-8), respectively. In addition, the reference antagonist has already been crystallized with the panda receptor as a more reliable model (PDB ID: 5U1X).

POINT 4: Page 2, line 53. The direction of ion flow is not only determined by the electrochemical gradient difference but also by the concentration difference.

Response 4: Thank you for your observation. We corrected the mistake.

Reviewer 2 Report

Comments and Suggestions for Authors

No laboratory-based investigation has been conducted to substantiate the theory, and the studies offered are insufficient in demonstrating the efficacy of genuine and optimal P2X7 antagonists.

Comments on the Quality of English Language

Typo and fluency issues are present on every page.

Author Response

Dear Reviewer,

We appreciate all comments and suggestions on the manuscript entitled “Combining shape-based and docking approaches for identifying potential P2X7 antagonist candidates from natural product libraries". We have considered all comments, as displayed below in a point-by-point response to each issue and/or comment. The text modifications are highlighted in the text. We hope that the manuscript is now publishable in the journal. 

Comments and Suggestions for Authors

No laboratory-based investigation has been conducted to substantiate the theory, and the studies offered are insufficient in demonstrating the efficacy of genuine and optimal P2X7 antagonists.

Response: Thank you for your comment. We completely agree that experimental validation is extremely important for confirming the predicted biological activities of compounds selected by virtual screening. We additionally agree that the lack of experimental validation is a limitation of our study. This point was discussed in several parts of our manuscript, as follows:

Abstract (last sentence, page 2, Revised Manuscript; also present in the Original Manuscript):

“(…) Therefore, the combination of shape-based screening and docking approaches applied in our study is proven useful in selecting potential novel P2X7 antagonist candidates from natural product-derived compound databases, which should be submitted to in vitro assays for the experimental validation of our VS.”

Introduction section (4th paragraph, page 3, Revised Manuscript; also present in the Original Manuscript):

“(…) Compounds selected by VS campaigns must, nevertheless, be submitted to confirmatory in vitro and/or in vivo assays for VS experimental validation [26,27,29–31].”

Introduction section (6th paragraph, page 4, Revised Manuscript; also present in the Original Manuscript):

“(…) In this regard, experimental validation steps are extremely relevant to overcome inherent docking limitations [31].”

Introduction section (last paragraph, page 4, Revised Manuscript; also present in the Original Manuscript):

“(…) The selected compounds represent structurally novel drug-like natural product-derived P2X7 antagonist candidates, which should be tested in in vitro assays for the experimental validation of our VS approach.”

Conclusions section (page 18, Revised Manuscript; also present in the Original Manuscript):

“(…) The compounds selected herein represent structurally novel and drug-like P2X7 antagonist candidates, which should be tested in in vitro assays for the experimental validation of our VS approach. (…)”

Additionally, in the revised version of the manuscript, we added the following sentences in the last paragraph of the Results and Discussion section, to discuss the importance of the experimental validation of VS models:

Results and Discussion section (last paragraph, page 16, Revised Manuscript):

“(…) Finally, we point out that these compounds should be tested in in vitro assays for the experimental validation of our VS. As mentioned in the Introduction section, the experimental validation is extremely important to confirm the utility of a VS approach due to the inherent limitations of in silico methods.”

Despite the limitation regarding in vitro assays, our intention with this work was to propose a new virtual screening model applied to the P2X7 receptor, since studies using this methodology are still quite scarce and very recent (doi: 10.15586/ijfs.v35i2.2288, 10.1111/cbdd.13867, 10.1016/j.compbiomed.2023.107299,10.3389/fphar.2022.1094607, 10.1016/j.bcp.2016.07.020). In this way, candidate compounds were identified based on the shape of the JNJ-47965567 antagonist, also presenting drug-like properties and, according to predictions, they are capable of interacting with key residues of the allosteric site of the P2X7 receptor. Therefore, the innovation that this methodology brings is to provide more robustness than conventional docking tests used in virtual screening.

Comments on the Quality of English Language

Typo and fluency issues are present on every page.

Response: Thanks for the observation. Grammatical errors were corrected by a native English speaker (The certificate is attached).

Reviewer 3 Report

Comments and Suggestions for Authors

The manuscript titled “Combining shape-based and docking approaches for selecting P2X7 antagonist 1 candidates from natural product libraries” submitted to Pharmaceuticals is an interesting study, and highlights the power of virtual screening. This was genuinely a pleasure to review, and the article should be published after a few minor changes.

1.       The title should be changed from “selecting P2X7 antagonist candidates” to “identifying potential P2X7 antagonist candidates” as the compounds identified have not been experimentally validated.

2.       Similar to above, the abstract should be altered from “proven useful to select promising novel P2X7 antagonist candidates” to “proven useful to select potential novel P2X7 antagonist candidates” as the compounds identified have not been experimentally validated.

3.       In lines 163, and 212 the authors highlight their selection method but reference “Figure 2” which depicts JNJ-47965567 – do the authors mean to reference Figure 1?

4.       The abbreviation AmP2X7 needs to be clarified.

5.       Line 197 references Karasawa, 2016 and states that all five antagonists require an interaction with “Asp-103”, but Karasawa state that residue 103 is a Phenylalanine. In the supplementary materials, the residue is also stated as F103. Can the authors please clarify.

6.       On line 223 the authors state that Phe-95 has been recognised as essential for binding affinity, but previously (above) discuss that binding to Asp-103 (potentially meant to be Phe-103), is the essential binding. Can the authors please clarify.

7.       In the Supplementary Material it states that the reproducibility of JNJ-47965567 is “No”, could the authors just clarify exactly what this means, and why this was chosen as the basis of the docking models?

8.       Finally, this study is an effective virtual screen of potential P2X7 antagonists, but could be broadly applied to other receptors/targets. The concluding sentence could be extended to highlight this. This is just a suggestion and can be ignored at authors discretion.

Author Response

Dear Reviewer,

We appreciate all comments and suggestions on the manuscript entitled “Combining shape-based and docking approaches for identifying potential P2X7 antagonist candidates from natural product libraries". We have considered all comments, as displayed below in a point-by-point response to each issue and/or comment. The text modifications are highlighted in the text. We hope that the manuscript is now publishable in the journal. 

Comments and Suggestions for Authors

The manuscript titled “Combining shape-based and docking approaches for selecting P2X7 antagonist 1 candidates from natural product libraries” submitted to Pharmaceuticals is an interesting study, and highlights the power of virtual screening. This was genuinely a pleasure to review, and the article should be published after a few minor changes.

POINT 1: The title should be changed from “selecting P2X7 antagonist candidates” to “identifying potential P2X7 antagonist candidates” as the compounds identified have not been experimentally validated.

Response 1: Thank you for your suggestion. We agree with your suggestion and change the manuscript title accordingly.

POINT 2: Similar to above, the abstract should be altered from “proven useful to select promising novel P2X7 antagonist candidates” to “proven useful to select potential novel P2X7 antagonist candidates” as the compounds identified have not been experimentally validated.

Response 2: Thank you for your suggestion. We agree with your suggestion and change the sentence accordingly.

POINT 3: In lines 163, and 212 the authors highlight their selection method but reference “Figure 2” which depicts JNJ-47965567 – do the authors mean to reference Figure 1?

Response 3: Thank you for your observation. We corrected this mistake and referenced it as Figure 1.

POINT 4: The abbreviation AmP2X7 needs to be clarified.

Response 4: Thank you for the observation. AmP2X7 refers to the Ailuropoda melanoleuca species (giant panda). We inserted this information in the manuscript.

POINT 5: Line 197 references Karasawa, 2016 and states that all five antagonists require an interaction with “Asp-103”, but Karasawa state that residue 103 is a Phenylalanine. In the supplementary materials, the residue is also stated as F103. Can the authors please clarify.

Response 5: Thank you for the observation. We corrected this mistake in the revised version of the manuscript. The correct is Phe-103.

POINT 6: On line 223 the authors state that Phe-95 has been recognised as essential for binding affinity, but previously (above) discuss that binding to Asp-103 (potentially meant to be Phe-103), is the essential binding. Can the authors please clarify.

Response 6: Thank you for your observation. We corrected this mistake in the revised version of the manuscript.

POINT 7: In the Supplementary Material it states that the reproducibility of JNJ-47965567 is “No”, could the authors just clarify exactly what this means, and why this was chosen as the basis of the docking models?

Response 7: Thank you for your question. The docking poses reproducibility was analyzed by visual inspection, considering the following criterion: the best-scored docking pose of each compound was considered “reproducible” if its conformation and orientation into the P2X7 allosteric binding site were superimposable to at least 4 other docking poses, among 10 docking poses. Based on this criterion, the following words were used to classify the docked compounds according to their best-scored docking pose reproducibility in the Supplementary Material tables: “yes” (the best-scored pose was considered “reproducible”), “no” (the best-scored pose was considered “not reproducible”), or “partially” (the best-scored pose was considered “partially reproducible”, which means that only part of the compound structure was superimposable with other docking poses).

Although the best-scored docking pose of JNJ-47965567 was not considered reproducible, many protein-ligand interactions observed in the crystal structure of the complex AmP2X7-JNJ-47965567 were reproduced by our docking procedures, as shown in the Supplementary Material tables.

We additionally stress that the 3D structure of the JNJ-47965567 antagonist from the crystal structure of the complex P2X7R-JNJ-47965567 (PDB ID: 5U1X, resolution: 3.20 Å), and not the docking pose of this compound, was used for building our shape-based model, as described in section 3.2 of the manuscript.

POINT 8: Finally, this study is an effective virtual screen of potential P2X7 antagonists, but could be broadly applied to other receptors/targets. The concluding sentence could be extended to highlight this. This is just a suggestion and can be ignored at authors discretion.

Response 8: We appreciated your suggestion and added a sentence in the conclusion section about this possibility.

Round 2

Reviewer 1 Report

Comments and Suggestions for Authors

The revised manuscript submitted by Alves et al. exhibits certain improvements over the previous version. However, it still does not meet the requirements for publication in this journal unless the authors show their functional analysis results.

Reviewer 2 Report

Comments and Suggestions for Authors

Thanks for response.